# Activity Measurement of ^44^Sc and Calibration of Activity Measurement Instruments on Production Sites and Clinics

**DOI:** 10.3390/molecules28031345

**Published:** 2023-01-31

**Authors:** Frederic Juget, Teresa Durán, Youcef Nedjadi, Zeynep Talip, Pascal V. Grundler, Chiara Favaretto, Pierluigi Casolaro, Gaia Dellepiane, Saverio Braccini, Claude Bailat, Nicholas P. van der Meulen

**Affiliations:** 1Institute of Radiation Physics, 1007 Lausanne, Switzerland; 2Center of Radiopharmaceutical Sciences ETH_PSI-USZ, Paul Scherrer Institute, 5232 Villigen-PSI, Switzerland; 3Albert Einstein Center for Fundamental Physics (AEC), Laboratory for High Energy Physics (LHEP), University of Bern, 3012 Bern, Switzerland; 4Laboratory of Radiochemistry, Paul Scherrer Institute, 5232 Villigen-PSI, Switzerland

**Keywords:** Sc-44 production, medical cyclotron, positron emission tomography, precise activity measurement, dose calibrator, γ-ray spectrometry, precise instruments calibration

## Abstract

^44^Sc is a promising radionuclide for positron emission tomography (PET) in nuclear medicine. As a part of the implementation of a production site for ^44^Sc, precise knowledge of the activity of the product is necessary. At the Paul Scherrer Institute (PSI) and the University of Bern (UniBE), ^44^Sc is produced by enriched ^44^CaO-target irradiation with a cyclotron. The two sites use different techniques for activity measurement, namely a dose calibrator at the PSI and a gamma-ray spectrometry system at UniBE and PSI. In this work, the ^44^Sc was produced at the PSI, and samples of the product were prepared in dedicated containers for onsite measurements at PSI, UniBE, and the Institute of Radiation Physics (IRA) in Lausanne for precise activity measurement using primary techniques and for the calibration of the reference ionization chambers. An accuracy of 1% was obtained for the activity measurement, allowing for a precise calibration of the dose calibrator and gamma-ray spectrometry of the two production sites. Each production site now has the capability of measuring ^44^Sc activity with an accuracy of 2%.

## 1. Introduction

Currently, there is a growing interest in scandium radioisotopes in nuclear medicine. In particular, ^44^Sc, which decays by β^+^ emission (94.27(5)%, end point = 1474.3(19) keV, average = 632.0(9) keV) and by electron capture (5.73(5)%) to excited states of stable ^44^Ca (Figure 1) [1], is a promising radionuclide for cancer diagnosis using positron emission tomography (PET) and positron emission tomography/computed tomography (PET/CT) [2,3] or for theranostics in combination with ^47^Sc [4].

For this study, the production of ^44^Sc was carried out using enriched Ca targets via the ^44^Ca(p,n)^44^Sc nuclear reaction by means of the research cyclotron at the Paul Scherrer Institute (PSI). When produced at a cyclotron, separation of ^44^Sc from ^44^Ca must be performed to yield a high radionuclidic and chemical purity. Due to the overlapping of the production cross section, a small amount (<1%) of ^44m^Sc is also produced and must be taken into account, as it decays into ^44^Sc (Figure 1). Moreover, very small amounts (<10^−5^) of other scandium radioisotopes (^46^Sc, ^47^Sc, and ^48^Sc) as well as ^88^Y can be detected due to the target composition. More details about target preparation, cross-section measurements, irradiation conditions, and chemical preparation realised at PSI and UniBE can be found in [5,6].

As part of the production of ^44^Sc at PSI and at the medical cyclotron of UniBE, the need to have a precise activity measurement of ^44^Sc sources has become pressing. In collaboration with the Institute of Radiation Physics (IRA), a standardization of a solution produced at PSI was performed in order to calibrate different systems used at PSI and UniBE to measure the activity and to calibrate the reference ionization chamber (CIR) at IRA. At PSI, the devices to determine the activity produced are a dose calibrator, where two glass vials filled with 5 mL and 1 mL, respectively, and a plastic Eppendorf filled at 1 mL are the measurement geometries, and a γ-ray spectrometry system using a plastic Eppendorf filled with 20 µL is also used to measure the amount of impurities. The same type of Eppendorf was sent to UniBE, where a γ-ray spectrometry system is used to measure the activity and impurities in the sample. The transportable reference ionization chamber (TCIR) from IRA [7] was moved to PSI in order to calculate a calibration factor using the glass vial filled at 1 mL. In addition to this calibration work, the same solution was used to measure the half-lives of ^44^Sc and ^44m^Sc, which are presented elsewhere [8].

This paper reports the impurity measurements using an HPGe detector, the results of the different techniques employed at IRA to standardize the ^44^Sc solution, and finally, the results of the calibration of the two ionization chambers at IRA, the dose calibrator and γ-ray spectrometry systems performed at PSI and UniBE with the standardized solution from IRA.

## 2. Materials and Methods

### 2.1. Source Preparation

An enriched ^44^CaO target (97%, Trace Sciences International, USA) was irradiated with ~11 MeV protons to exploit the ^44^Ca(p,n)^44^Sc nuclear reaction. ^44^Sc was separated and concentrated using extraction and ion exchange resins [3]. All these procedures were performed at PSI, and the solution, consisting of 0.1 M HCl, was used to prepare different samples. Two glass penicillin vials (one of them sterile) and one plastic Eppendorf, each filled with 1 mL solution, and one plastic Eppendorf filled with 0.02 mL were prepared for measurement at PSI. The plastic Eppendorf filled with 0.02 mL was sent to UniBE, and the remainder of the solution, approximately 2 mL in a glass vial, was sent to IRA for a primary standardisation. Each container was weighed before and after filling in order to determine the precise radioactive mass, reported in Table 1. The containers measured at PSI and UniBE were planned for the calibration of the dose calibrators and γ-ray spectrometry systems using the activity value measured at IRA. After the filling of each container at PSI, the samples were immediately transported to UniBE and IRA to avoid too much of a loss of activity due to the ^44^Sc decay since its half-life is only four hours. The samples were produced and sent on 19 November 2021, so the reference time for all measurements was 19 November 2021, 12:00 UTC.

The sample delivered to IRA was diluted to prepare several sources for the activity measurements. Figure 2 shows the scheme of solution handling and source preparation and their use. The solution was first diluted by a factor F_dil_ = 10.959(1) and used to prepare sources for activity measurement using 4πβ(PS)-γ coincidence method [9,10] and sealed ampoules with 3 g of solution for calibration of IRA reference ionisation chamber, which is traceable to the SIR, the international reference system for the Becquerel [11]. This solution was again diluted by a factor (F_dil_) ~3.8949(3) to prepare two vials to be measured on a high-energy resolution HPGe detector to determine and quantify the presence of impurities. Additionally, several vials were prepared with this second solution to measure the activity using two liquid scintillation techniques, namely TDCR (the triple-to-double-coincidence ratio) and CIEMAT/NIST [12], as well as for a second set of 4πβ(PS)-γ coincidence sources for corroborating measurements.

### 2.2. Impurity Evaluation at IRA

The impurity evaluation was performed by γ-ray spectrometry with a high-energy resolution HPGe detector. The two polyethylene vials containing the ^44^Sc solution were measured over a period of 10 h to assess the activity of the ^44m^Sc impurity.

In the spectrum shown in Figure 3, it is possible to observe the ^44m^Sc lines, and its activity was calculated using the 271 keV line with 86.7(3)% intensity [13]. The ^44^Sc activity was calculated using the line at 1499 keV with 0.908(15)% intensity. Therefore, the activity ratio ^44m^Sc/^44^Sc at the reference time, calculated using the Bateman equation and the half-lives of 4.042(3) h and 58.6(1) h for ^44^Sc [14] and ^44m^Sc [15], respectively, was 0.00756(14). This quantity is not negligible and will be taken into account for the ^44^Sc activity calculation.

An additional γ-ray spectrometry measurement was performed seventeen days after production, using the mother solution produced at PSI, to check whether long-lived radionuclides were present in the solution after the decay of ^44^Sc. Small quantities (~10^−5^) of ^46^Sc, ^47^Sc, ^48^Sc, and ^88^Y were found, resulting from the activation of impurities in the target material [5]. Table 2 presents the activity ratios of the identified radionuclidic impurities found. As they were present in small quantities and did not have an impact on the measurements, they were neglected for the ^44^Sc activity calculation.

### 2.3. Standardisation Methods

Both dilutions from the original solution prepared at PSI were measured by means of various activity standardisation techniques.

#### 2.3.1. 4πβ(PS)-4πγ(NaI) Coincidence Counting

The β-detector in this system [16,17] consists of a plastic scintillator with a dry radioactive deposit at its centre, allowing full solid angle geometry. The scintillator is coupled to the window of a Photonis XP3132 low-noise PMT that is inserted into the well of a Quartz & Silice 127-SPE-127 12.5 cm × 12.5 cm NaI(Tl) detector. In this configuration, the radioactive source is placed at the bottom of the well, interacting with the γ-ray detector through 99.1% of the full solid angle. The scintillator crystal and the photomultiplier (RTC type XP2050) are housed within a cylindrical shielding. Analogue electronics were used to perform the signal processing, and the efficiency variation was achieved by electronic discrimination. A deadtime of 29 µs was used in both β- and γ-channels, and the coincidence window was set at 1 µs.

One source from the concentrated dilution was measured in two γ-energy regimes, setting a threshold at 364.5 keV and another at 661.7 keV. A source from the second dilution was measured with the threshold set at 364.5 keV only. When taking into account the presence of the ^44m^Sc impurity and its non-equilibrium with the ^44^Sc that it generates, the extrapolation curves of the ^44^Sc activity concentrations were found to be linear in all cases and yielded values compatible within 0.3%.

#### 2.3.2. Liquid Scintillation Counting

To perform liquid scintillation measurements, aliquots of the less-concentrated solution were dispensed gravimetrically into vials prefilled with 14.5 mL Ultima Gold cocktail and topped up with various volumes of ultrapure water to achieve a 6.5 % aqueous fraction.

The TDCR method along with the CIEMAT-NIST (CNET) technique were used. Three vials were measured with the TDCR system [8] using a MAC3-module [9,10] with 100 ns resolving time. Voltage defocusing varying from 560 and 340 V in 40 V decrements was applied to change triple- and double-coincidence detection efficiencies. Three other vials were also measured with the IRA commercial TriCarb 2700 LS counter, along with eleven quenched tritium samples to determine the calibration curve for the CNET method.

Since the source contains a ^44m^Sc impurity that engenders ^44^Sc, the double detection efficiency in the TDCR or the TriCarb counter reads:$$\varepsilon_{D}=\varepsilon_{D44}\cdot \left[1+\theta \cdot f\left(\lambda_{44m},\lambda_{44}\right)\right]+\alpha_{44-44m}\cdot \theta \cdot f\left(\lambda_{44m},\lambda_{44}\right)\cdot \varepsilon_{D44m}$$
where *α*_44-44m_ is the ^44m^Sc/^44^Sc equilibrium factor, *θ* is the ^44m^Sc/^44^Sc activity ratio, and 𝑓(λ_44𝑚_, λ_44_) is the ^44m^Sc/^44^Sc decay factor ratio.

The TDCR triple and double detection efficiencies and the CNET double detection efficiencies were calculated with a Fortran code, developed at IRA, which takes the PMTs asymmetry and the micelle effect into account [18]. This code computes the efficiencies using the stochastic approach, in which the multimillion sampled energies of the β–particles, the conversion, Auger, and photoelectrons produced in a vial are obtained from Geant 4 Monte Carlo simulations involving the Radioactivity and Atomic relaxation modules. The nuclear and atomic data used are from DDEP and ENSDF [19,20].

### 2.4. Reference Ionisation Chambers

The CIR chamber is calibrated using primary measurement techniques for more than 30 isotopes and is traceable to the SIR for the activity unit, the Becquerel. It has been in operation since 1983 and is regularly maintained and checked to ensure its stability [21]. The measurements with the CIR are regularly validated using international comparison in the framework of the SIR [22,23]. Radioactive solutions, precisely characterized using primary techniques, are used to fill a dedicated ampoule filled with 3 g and accurately weighed with a precision balance. The ampoule is used to precisely calibrate the CIR chamber, and usually, its calibration factor, called equivalent activity Ae, has an accuracy of approximately 0.5%. Once Ae is known, the CIR is used to measure the unknown activity of any solution containing the radionuclide of interest. The chamber is used to measure the activity of solutions and to produce standards that are traceable to the SIR. In Switzerland, the Swiss Federal Office of Metrology (METAS) has designated IRA as the institute for radioactivity measurement.

The two ^44^Sc ampoules (Figure 2) were centrifuged, for optimal efficiency and reproducibility, before measurement in the CIR. A measurement cycle is the time needed to load a given capacitor up to 0.1 Volt six times with the charge produced by the ionisation in the chamber. The ionisation current is given by the mean and its standard deviation obtained from 20 consecutive reading cycles of the current. A measurement sequence starts with a background measurement using a small 503 pF capacitor. A typical background value is 0.057(2) pA. For the stability check, two reference ^137^Cs sources were first measured with a larger capacitor (30,090 pF), after which the ^44^Sc ampoules were measured with the same capacitor. Finally, another background measurement was performed with the small capacitor to complete the sequence. The results were used to calculate a CIR calibration factor for ^44^Sc using the activity measured with the primary techniques described above.

The TCIR is an ionization chamber which is designed to measure short-lived radionuclides and is easily transportable to production sites or nuclear medicine centers. It is traceable to the CIR, as it is calibrated using solution measured with the CIR and, therefore, with a known activity with an uncertainty of approximately 0.5%. The measurement with the TCIR chamber is less precise than that obtained with the CIR, and the uncertainty on the calibration factor is usually approximately 1% or more [7]. Recently, its measurement geometry was adjusted to a 5 mL sterile vial filled to 1 mL, instead of a 10 mL vial filled to 5 mL, to facilitate the preparation of the measurement sample. Therefore, taking advantage of the need to calibrate the dose calibrator at PSI for the sterile vial filled at 1 mL, the TCIR was also calibrated for ^44^Sc.

### 2.5. Instruments Calibration at PSI

The sterile glass vial, the penicillin vial, and the Eppendorf filled at 1 mL were measured using the dose calibrator ISOMED 2010 [24]. The detector is a well-type ionisation chamber connected to an embedded acquisition system and interfaced with a PC running under Windows with a dedicated software. The software using the calibration coefficient directly provides the activity value in Bq. The value of 0.1045 as the calibration factor for ^44^Sc was already implemented in the software by the manufacturer. The activity of two vials and the Eppendorf were measured using this value. The values obtained were compared with that expected from the primary measurement technique. The deviation obtained was used to correct the calibration factor for the different containers. The effect of the ^44m^Sc activity was assessed in order to consider whether it ought to be taken into account.

### 2.6. Instruments Calibration at UniBE

The Eppendorf vial containing the scandium solution was measured with the HPGe detector in operation at the UniBE cyclotron laboratory. The detector (Canberra GR2009) is an N-type coaxial HPGe with the sensitive volume shielded by 10 cm of lead. The preamplifier signal is sent to a Lynx^®^ digital analyser featuring the Genie2K analysis software and the Microsoft Excel application Excel2Genie [25]. The Eppendorf was placed in a vertical position on the detector by means of a custom holder. In this condition, the vial could be considered to be a point-like source.

The distance with respect to the crystal was adjusted over time according to the activity of the source in order not to saturate the detector. For this purpose, an aluminium stand as well as an acrylic–glass stand were used, which allowed the source to be positioned up to 100 cm and 10 cm (Figure 4) away from the detector, respectively.

## 3. Results

### 3.1. Standardisation Results

Results of the primary measurements are summarised in Figure 5. The 4πβ-(PS)-4πγ(NaI) coincidence counting measurements were clearly consistent across γ−energy regime and activity concentration. The liquid scintillation methods (TDCR and CNET) also agreed to within 0.9%.

These two sets of techniques predicted close ^44^Sc activity concentrations, as their averages deviated from each other by only 0.1%. Nevertheless, the adopted average activity concentration value was that predicted by 4πβ(PS)-4πγ(NaI) coincidence counting, i.e., 41.30 ± 0.41 MBq at the reference date (19.11.2021 12:00 UTC). This result was independent of the ^44^Sc or ^44m^Sc decay data.

### 3.2. Reference Ionisation Chambers Measurements

The measurements of the currents produced in the CIR were corrected for the contribution of the ^44m^Sc impurity using a semi-empirical calibration factor obtained from the response curve of the CIR chamber [26]. This contribution is approximately 0.3%. The current per mass value measured for the two ampoules was 346.60 pA/g and 347.02 pA/g, at the reference date, respectively, providing good agreement, lower than 0.2%. Using the average current between the two ampoules and the activity obtained from the primary measurements, the calibration factor could be calculated for the CIR. It was determined that Ae = 8.176 ± 0.89 (1.1%) MBq.

The current measured for the sterile vial with the TCIR was also corrected for the contribution of the ^44m^Sc impurity using the response curve of the TCIR [26]. The current per mass value was 2751.48 pA/g at the reference date. This value corresponds to the activity of the mother solution and differs from that obtained previously with the CIR. The difference is explained not only by the dilution of the mother solution to have enough volume to fill the different geometries used for the primary measurement techniques and CIR measurement at IRA (Figure 2) but also by the different geometries used (ampoule and vial) as well as the type of the ionization chamber, which have different well size and gas chamber volume. Using the current measured and the activity obtained from the primary measurements, the calibration factor for the TCIR was determined to be 15.07 ± 0.17 (1.1%) MBq/pA.

### 3.3. Measurements with Dose Calibrator at PSI

The activity measurements for the sterile vials using the provider calibration factor were 45.48 MBq/g, 45.40 MBq/g, and 44.65 MBq/g for the Eppendorf, at the reference date. Since the difference in activity measured for the two glass vials was small (~0.2%), a single coefficient could be used for calibration. Using the relation provided by the manufacturer, the coefficient was calculated to be 0.0950. For the Eppendorf, the difference to the glass vials is almost 2%; therefore, a specific coefficient was determined to be 0.0967.

It is worth mentioning here that no correction for ^44m^Sc contribution was made. A calculation using the Bateman equation and an activity ratio ^44m^Sc/^44^Sc of 1% at the production time (target irradiation) will provide an overestimate of less than 1% of the activity measured with the dose calibrator for up to 12 h after the irradiation time. This is also due to the response of the ionisation chamber, which is approximately seven times lower for ^44m^Sc than for ^44^Sc. For other types of ionisation chambers, the response may be slightly different and should be checked independently. In this case, the activity ratio was calculated to be 0.4% at the end of irradiation (EOI) and would provide an overestimation of less than 0.4% after 12 h, which is within the uncertainty provided by the dose calibrator measurement. Therefore, if the impurity ratio is below 1% at EOI, the presence of ^44m^Sc can be neglected for measurements with dose calibrators if the measurement is performed within 12 h after the EOI. This corresponds to approximately three times the ^44^Sc half-life and is larger than the time foreseen for medical application.

### 3.4. Measurements with γ-Ray Spectrometry System at UniBE

For gamma spectrometry, each γ-line can be separated; therefore, the activity calculation was performed by considering the 1499 keV and 271 keV lines for ^44^Sc and ^44m^Sc, respectively, and using the Bateman equation. The results are presented in Table 3 at the reference time (19 November 2021, 12:00 UTC), showing a compatibility within 2.2% of the reference value. Therefore, no correction on the calibration factor is required.

## 4. Conclusions and Outlook

The production of ^44^Sc for clinical applications in nuclear medicine requires an accurate measurement of the activity for each sample delivered, and a specific methodology was developed and tested. The ^44^Sc was produced using the research cyclotron at PSI, and a precise measurement of its activity was performed at IRA using primary standardization techniques, providing an accuracy of 1%. The presence of the Sc impurities, mainly a ^44m^Sc with an activity ratio of approximately 0.4% at EOI, was measured with a γ-ray spectrometry system and was taken into account in the calculation. The measurements showed that if the ^44m^Sc ratio is below 1% at EOI, its presence can be neglected for measurements with dose calibrators.

This precise knowledge of the activity allowed for the calibration of the dose calibrator at PSI for two sample geometries and the ability to check the calibration of the γ-ray spectrometry system at UniBE. The latter system indicated a difference of 2.2%, which confirmed the validity of its calibration; therefore, no modification was required. The TCIR chamber was also calibrated for ^44^Sc. This is an important achievement, as this chamber is intended to be moved to production sites or nuclear medicine centers where it can be used to measure samples with an uncertainty lower than 2% and allow for the calibration of activity measurement instruments on site.

Due to the short half-life of ^44^Sc, all the activities, namely production, logistics (from PSI to Bern and Lausanne), and measurements, were realized in approximately half a day. ^44^Sc must be produced relatively close to the nuclear medicine centers to where it is shipped for administration to patients. The requirement is to have a properly calibrated instrument for γ-spectrometry at the production site in order to measure the activity and any impurity as well as to calibrate the instruments for activity measurement before injection into patients.

This methodology is new for ^44^Sc and necessary in view of potential clinical trials. It can be applied at any ^44^Sc production site as well as nuclear medicine centers using this novel radionuclide. The findings reported in this paper contribute to paving the way towards the use of ^44^Sc for theranostics in nuclear medicine.

## Figures and Tables

**Figure 1 molecules-28-01345-f001:**
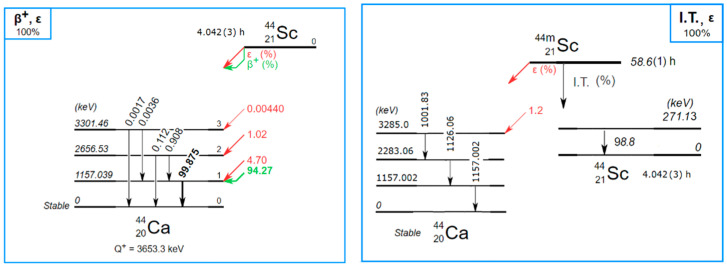
Decay scheme of ^44^Sc and ^44m^Sc. Data from [1].

**Figure 2 molecules-28-01345-f002:**
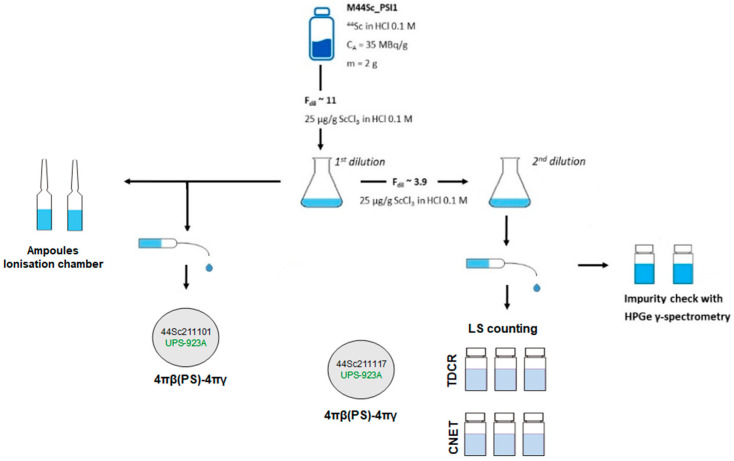
Source preparation scheme from the solution received from PSI on 19 November 2021, used for the measurements at IRA.

**Figure 3 molecules-28-01345-f003:**
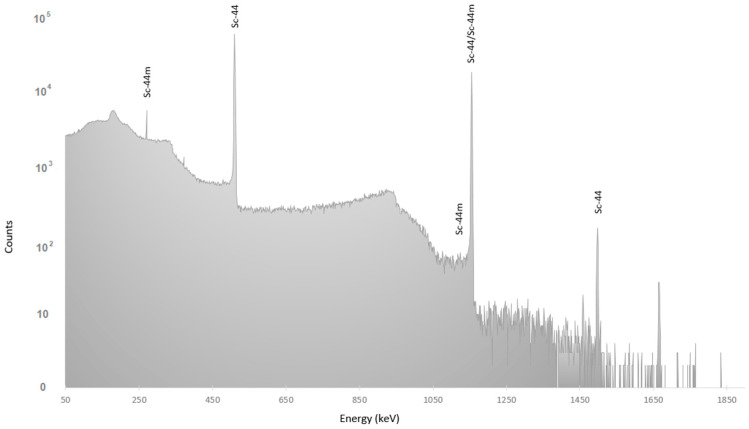
γ-ray spectrum measured on 19/11/2021 with the HPGe to quantify the ^44m^Sc impurity.

**Figure 4 molecules-28-01345-f004:**
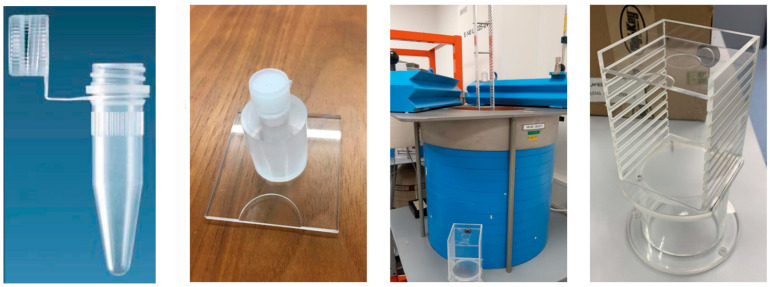
Images of the Eppendorf positioning system at the University of Bern. From left to right: Eppendorf vial, Eppendorf holder, detector enclosure, and positioning device.

**Figure 5 molecules-28-01345-f005:**
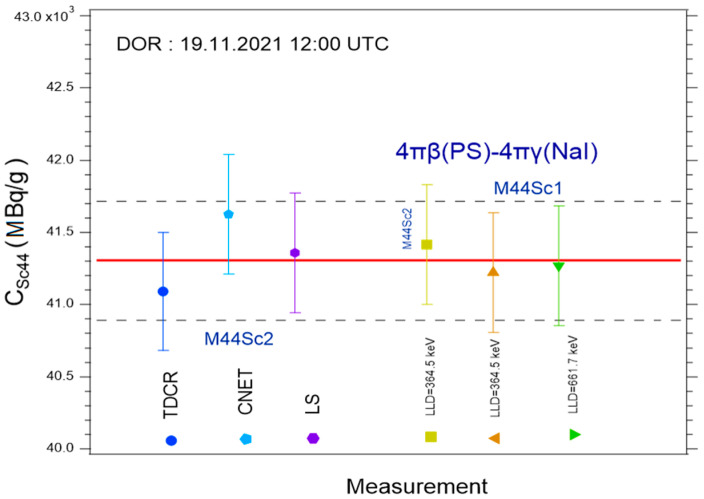
Results of the activity measurement of the mother solution using various primary techniques at IRA.

**Table 1 molecules-28-01345-t001:** The mass of the radioactive solution (1 g unit) used to fill the different containers.

	Sterile Glass Vial (g)	Eppendorf Vial (g)	Penicillin Vial (g) (Made up to 5 g with Cold Solution)	Eppendorf (g) (Sent to UniBE)
Empty	10.6028	1.4426	15.9722	1.4352
Filled	11.6067	2.4453	16.9725	1.4530
Rad. mass (g)	**1.0039(40)**	**1.0027(8)**	**1.0003(4)**	**0.0178(9)**

**Table 2 molecules-28-01345-t002:** Impurities activity ratios with respect to ^44^Sc at the reference time.

	Ratio	Unc.	Rel. Unc. (%)
**Sc-47/Sc-44**	**1.050 × 10^−5^**	2.6 × 10^−7^	2.5
**Sc-48/Sc-44**	**5.716 × 10^−5^**	1.73 × 10^−6^	3.0
**Sc-46/Sc-44**	**4 × 10^−7^**	2 × 10^−9^	4.8
**Y-88/Sc-44**	**3.51 × 10^−6^**	7 × 10^−8^	1.9

**Table 3 molecules-28-01345-t003:** Activity measurement using the γ-ray spectrometry system at UniBE.

	UniBE A (MBq/g)	Difference with Reference
Sc-44	42.2 ± 0.3	2.2%
Sc-44m	0.299 ± 0.002	

## Data Availability

Data will be made available on request.

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
