# Peer review of "Activity Measurement of 44Sc and Calibration of Activity Measurement Instruments on Production Sites and Clinics"

_molecules, 2023, doi:10.3390/molecules28031345_

Round 1

Reviewer 1 Report

The paper is well written; however, it is not clear to me what the added value of the research is for the international scientific community (in particular, the originality and novelty of the work presented is not adequately highlighted).

Author Response

Reviewer:

The paper is well written; however, it is not clear to me what the added value of the research is for the international scientific community (in particular, the originality and novelty of the work presented is not adequately highlighted).

Response:

What is shown in this study is the possibility to use of Sc-44 from the cyclotron despite it contains impurity of Sc-44m whose presence can be neglected, if it is below 1% at the production time, when the activity is measured with a dose calibrator. The dose calibrator is an easy-to-use instrument, intensively used in nuclear medicine centres to measure accurately the sample activity. The presence of impurity in the case of Sc-44 must be low otherwise the direct measurement in a dose calibrator is wrong. Here we show both, the production of low-impurity Sc-44 and the correct measurement of the activity with a dose calibrator despite the presence of this low contribution of Sc-44m.

This is highlighted in the conclusion and how the different instruments are calibrated with a traceability to a reference system of the Becquerel Unit using the same Sc-44 solution, which is not usually done like this for short half-life isotopes i.e it is not done on-site.

Reviewer 2 Report

I am grateful for the opportunity to review manuscript with ID: molecules-2141687, entitled “Activity measurements of 44Sc and calibration of activity measurement instrumentation on production sites and clinics.

The authors correctly point out the growing interest in 44Sc as a PET or PET/CT tracer for nuclear medicine applications. They go on to describe their study of comparative activity measurements at a production site and a clinical facility, benchmarked to a traceable and standardized calibration at a national standards lab.

This work is of interest to the nuclear medicine physics community and provides significant insight. This work is therefore a useful addition to the literature. However, I do have one comment:

·        There are no data for liquid scintillation coincidence counting as “unforeseen stability issues” with the electronics rendered these data unusable. Why were these issues not resolved and data collection repeated? Would the authors either solve these issues and repeat all measurements as necessary or provide a thorough discussion of the impact the omission of these data have on the overall paper.

Author Response

Reviewer:

There are no data for liquid scintillation coincidence counting as “unforeseen stability issues” with the electronics rendered these data unusable. Why were these issues not resolved and data collection repeated? Would the authors either solve these issues and repeat all measurements as necessary or provide a thorough discussion of the impact the omission of these data have on the overall paper.

Response: We had some efficiency loss problems with the data acquisition electronics that made unavailable the efficiency extrapolation that is performed when using the coincidence counting system. The measurement was performed on November 2021 and the data analysis on January 2022, so by the time we discovered the issue it was too late to repeat the data collection as the source had decayed away.

The impact of the omission of these data is not very significant, as we already had enough results from independent primary measurement methods to perform a consistent calculation of the reference activity concentration.

Finally, we removed all the text referencing this measurement and the figure 2 was modified accordingly.

Reviewer 3 Report

The authors describe the method of Sc-44m impurity activity measurement in production environment of Sc-44, a radionuclide for Positron Emission Tomography and theranostic with Sc-47 treatment.

Major

1. Gamma Sc-44 emission at 1157 keV is close to double 511 keV (=1022 keV). For HPGE detector, this is not an issue, but if NaI(Tl) or similar low energy detectors are used, how is this close emissions (13%) resolved or how does double 511 keV peak is accounted for?

2. If the authors analyze for presence of Sc-44m, why not to present its level scheme in Figure 1 as a second scheme? 

3. Why was Na-22 or similar well-known beta+ and gamma source not used for calibration and only Cs-137? 

4. In line 113 the activity ratio of Sc-44m/Sc-44 is 0.00756. Multiplying by half-lives, we obtain that Sc-44m isotope concentration is about 0,11 of Sc-44, which is an important factor, mentioned in line 114. Then in line 292 the ratio is only 0.4%. I fail to understand this discrepancy.  Moreover the 11% contamination by unwanted isotope, rises the security concerns for the future patients.

5. If  "4πβ(LS)-γ(NaI) coincidence counting" system was not used due to failure, why describe it? If the results expected from it are so important, please repeat Sc-44 production.

Minor

1. References are in a format which is not MDPI. They lack DOI numbers, which makes the search difficult. 

2. I do not have access to Ref. 2. Please provide beta+ positron maximum energy, or if there is more than one emission (I doubt), provide median and maximum. Provide distance in water, as compared to the same parameter for F-18. These are important characteristics for PET imaging. I may look them up in the web search engine, but a table or data would increase influence of the article. You can refer to the article http://dx.doi.org/10.1016/j.apradiso.2016.01.006 where PET image quality is compared (I´m not an author of that publication). 

3. Line 292: Please mention that it is concentration of activity and not an isotope concentration.  

4. Half-lives of Sc-44 in Figure 1 and in line 113 are different. I understand that half-life of Sc-44 is debated, but either correct or explain this discrepancy.

Reviewer 4 Report

A very interesting and useful paper. The use of language and style is excellent. I do have a few suggestions which would in my mind improve the paper and make it more accessible to a wider audience. 

- it would be good to mention the beam already in the abstract. It is assumed the reader knows it's a proton beam. Also, the beam parameters are not specified, though relevant as the overlap of the beam energy distribution with the production cross section of 44mSc and 44Sc is most relevant to the study. Come to think about it, it would be worthwhile to compare a beam related estimate with the experimental findings.

- figure 1 is missing the bracing ratios in the decay of 44Sc. Also, the text is missing a treatment of the electron capture branch.

- experimental details of the apparatus are scarce and sometimes seem to be contradictory (e.g. the volumes on page 2 top and bottom). Also, the solution is not described, nor is the travel time. Uncertainties e.g. on the dilution factor are not given.

- I quite like figure 2, all the more baffling that detector geometries are lacking. It would certainly be easier to understand (and also clarify the use of two different PMTs) if they were included. Similarly, no details or results of the calibration are given.

- it would be beneficial to compare to the clinically required precision.

Please give some thought to estimate a potential 44mSc contamination based on the beam parameters and how this would compare to your measurements.

Round 2

Reviewer 1 Report

In my opinion the manuscript can be accepted if the considerations reported in the author's reply will be more explicitly reported into the final text (further highlighting the originality and the contribution for the scientific community of the performed research).

Author Response

New sentences were added in the conclusion as requested by the reviewer.

Reviewer 2 Report

The authors have adequately addressed reviewer comments with this revised submission.

Author Response

According to the reviewer the comments were addressed adequately.